# Preconception mental health (Healthy Life Trajectories Initiative): Identifying factors associated with probable anxiety and depression among young women living in urban-poor South Africa

**Shane A. Norris**[1,2]*, **Claire Hart**[1], **Lukhanyo H. Nyati**[1,3], **Wihan Taljaard**[1], **Rayjean J. Hung**[4,5], **Ravi Retnakaran**[4,5], **Stephen Lye**[4,5], **Catherine E. Draper**[1], **Ashleigh Craig**[1]

**1** SAMRC/Wits Developmental Pathways for Health Research Unit, Department of Pediatrics, Faculty of Health Sciences, University of the Witwatersrand, Johannesburg, South Africa, **2** School of Human Development and Health, Faculty of Medicine, University of Southampton, Southampton, United Kingdom, **3** Interprofessional Education Unit, Faculty of Community and Health Sciences, University of the Western Cape, Bellville, South Africa, **4** Sinai Health System, Mont Sinai Hospital, Toronto, Canada, **5** Department of Medicine, University of Toronto, Toronto, Canada

* s.a.norris@soton.ac.uk

## Abstract

Mental health disorders affect millions worldwide, with socially vulnerable youth in urban environments disproportionately affected. South Africa (SA) remains one of the most inequitable countries, and specific pathways linking poverty to mental health remains unclear. This cross-sectional study analysed baseline data from the Bukhali trial in Soweto, SA part of the global Healthy Life Trajectories Initiative (HeLTI). Young women (n = 7735) completed surveys with physical assessments covering sociodemographic, household-level and behavioural-level factors, and mental health. Among the women, 12.6% reported anxiety, 15.8% reported depression, and 9.7% experienced both. Hazardous alcohol use (20.0%) and poor sleep (55.5%) were commonly reportedly behavioural factors among these women. Being in a committed relationship reduced the odds of reporting anxiety and depression (OR ≥0.66), while childhood adversity, hazardous alcohol use, and poor sleep increased the odds (OR ≥1.29). Moderate to severe anxiety greatly increased the risk of depression (OR 32.20). In the comorbid model, while being in a committed relationship remained protective (OR 0.67), childhood adversity and poor sleep quality was associated with substantial risk (OR ≥1.31) of this co-morbidity. In a gSEM constructed *a priori*, significant direct associations were found for poverty (measured by household socio-economic status) on alcohol use (*p* = 0.015), childhood adversity on mental health (*p* < 0.001), and committed relationship on anxiety (*p* < 0.001). Mediation analysis revealed that poverty affected anxiety partially through poor sleep (54.2%), and fully via depression (86.9%), and affected depression fully via sleep (43.7%). Childhood

**Data availability statement:** All data included in the analyses for this paper, and accompanying codebook, are publicly available on the HeLTI website (https://helti-hub.tghn.org/cohorts/south-africa/). Access to the de-identified participant-level dataset is available upon request to the corresponding author. Requests should include the requestor's contact details, institutional affiliation, and research question(s).

**Funding:** This work was supported by the South African Medical Research Council (SAMRC) (no reference number to SAN) and the Canadian Institutes of Health Research (ISA-181192 to SL). The funders had no role in study design, data collection and analysis, decision to publish, or preparation of the manuscript. The authors received no specific funding for this work, except for CED who is funded from the SAMRC grant.

**Competing interests:** The authors have declared that no competing interests exist.

adversity associated with depression partially via anxiety (79.4%), sleep (31.3%), and alcohol use (14.2%), and anxiety through depression (88.6%) and sleep (42.2%). In conclusion, childhood adversity, poverty and behavioural factors co-occur, and are collectively associated with elevated symptoms of anxiety and depression among young women in urban-poor settings. While tackling structural inequalities is critical, strengthening mental health support networks in these settings could aid women.

## Introduction

Mental health disorders, particularly anxiety and depression, rank among the leading causes of disability worldwide, affecting over 970 million individuals and contribute substantially to the Global Burden of Diseases [1,2]. Particularly, socially vulnerable younger people are disproportionately affected by poorer mental health, especially those living in urban-poor environments and who have experienced childhood trauma [3,4]. Gaining a more comprehensive understanding of factors and pathways that are associated with increased risk of common mental health conditions in vulnerable populations is imperative for effective primary prevention and mental health support strategies.

While South Africa has endeavoured to dismantle the legacy of apartheid, it remains one of the most unequal countries globally (Gini coefficient of 0.63). The combination of persisting spatial and economic inequalities, particularly living in historically disadvantaged 'townships', traps many families in a cycle of poverty with greater risk of psychological distress [5]. In nationally representative South African panel studies, data revealed high levels of childhood adversity, probable anxiety, depression and associations with food insecurity. During the COVID-19 pandemic (in 2021), the national prevalence of household food insecurity was at 20.4%, mean Adverse Childhood Experience (ACE) was 2 events (with standard deviation (SD) of 2.6), and the adult prevalences of reported symptoms of anxiety and depression were 17.8% and 25.7% respectively [6,7]. Shifting from 'food secure' to 'at risk' or from 'at risk' to 'food insecure' group was associated with 1.7 times greater odds of being in a higher category of anxiety or depression ($p < 0.001$) [8]. These prevalences remained unchanged in the repeated 2022 (late COVID-19) national panel study, except that food insecurity in households with children was 23.7% [9,10].

Although the association between poverty and common mental conditions are well documented, the direct and indirect pathways through which socioeconomic status (SES) affects anxiety and depression within urban African contexts are not fully elucidated. Informed by a socioecological framework, household SES (exposure) exerts both direct and indirect associations on anxiety and depression through key mediators, such as household- (e.g., food insecurity) and individual-level factors (e.g. ACEs and health behaviour). This framework acknowledges the layered and intersecting nature of social disadvantage in shaping mental health pathways. Therefore, our study aims were two-fold: (i) investigate the direct and indirect association of household-level SES and probable anxiety and depression among young women

aged 18–28 years in an urban-poor community (Soweto, South Africa); and (ii) determine the extent to which individual-level factors mediate the relationship between SES and anxiety and depression. We acknowledge that there are social and structural levers shaped by South Africa's apartheid legacy of spatial segregation, educational and employment exclusion, and gendered violence that contribute to anxiety and depression risk, and that this is beyond the scope of the current paper. This study is restricted to one historically under-resourced urban area (Soweto), consequently many macro-level apartheid-structured factors (province, urbanicity, service regimes, legacy spatial planning) are held constant, which improves internal validity versus a multi-site sample. This is a strength for the trial study component, but within Soweto there's still likely micro-structural variation (ward/section differences in housing density, transport access, lighting/safety, policing/crime, school quality, clinic proximity, crowding) and these could impact associations.

## Methodology

### Ethics statement

All procedures were reviewed and approved by the Human Ethics Research Committee of the University of the Witwatersrand, Johannesburg, South Africa (M171137, M1811111), the Mount Sinai Hospital, Canada (19–0066-E) and the WHO Ethics Committee (ERC.0003328). The trial was also registered with the Pan African Clinical Trial Registry (PACTR201903750173871). Written informed consent was obtained from all participants. Participants identified as needing mental health support were referred to local health services.

### Study design and recruitment

This cross-sectional study used data from the baseline (prior to randomisation) of the *Bukhali* trial (Soweto, South Africa) [11], which is part of the Healthy Life Trajectories Initiative (HeLTI). HeLTI is a consortium of four harmonised randomised controlled trials in Canada, China, India and South Africa, which aims to evaluate a continuum of care intervention that is initiated in preconception and for those who become pregnant, continues during pregnancy and postpartum. The intervention package is designed to optimise physical and mental health of women to improve their current and future health and offset intergenerational obesity risk.

Urban Soweto has a population of over 1.4 million people and is characterised by high levels of social vulnerability, [6] making it a pertinent setting for examining the socioeconomic association on young women's mental health. To maximise representativeness, from the entire area of Soweto using k-means clustering, we defined thirty random community clusters with a 1km$^2$ radius each to minimise the sum of squares within each cluster [12]. From these clusters, we randomly approached 38634 households, of which 17272 households had young women occupants (aged 18–28 years). Women were ineligible for recruitment if they were recently diagnosed with any cancer, type 1 diabetes and/or epilepsy given their care needs, or presented with cognitive impairment that could influence their ability to provide informed consent. From these households, 12402 eligible women were able to be contacted and deemed eligible for enrolment into the study, and of these, 10592 women consented to participate. Consented women were then invited to our research centre within the Chris Hani Baragwanath Academic Hospital precinct in Soweto for an interviewer-administered baseline survey with physical measurements and biological sample collection. Of those that consented, 7735 women attended their data collection booking visit within the timeframe at the research centre (refer to **S1 Fig** for the consort diagram). Data collection was conducted from May 2019 to June 2023, with a temporary suspension of recruitment during 2020 due to the COVID-19 pandemic.

### Data collection procedures

Trained research assistants conducted structured face-to-face interviews in private rooms at the research centre at Chris Hani Baragwanath Academic Hospital. Data were collected using pre-coded, interviewer-administered questionnaires on

tablets using REDCap. All interviews were conducted in English by multi-lingual staff that could converse or further explain the question in a local language if needed; instruments were translated and piloted prior to use.

**Sociodemographic factors.** Participants provided demographic information, including age, marital status (single, married, cohabiting, separated/divorced), highest level of education (no schooling, primary, secondary, tertiary), current employment status (employed, unemployed, student), and receipt of government social support (child support, disability, or unemployment grants). Parity was determined by asking whether participants had ever given birth and, if so, how many children they had. A household asset score, an indicator of household SES, was computed and used as an indicator of economic differentiation [13] and included a tally of thirteen household amenities. Information on household composition and living conditions was also collected, including the total number of household members and the number of rooms used for sleeping, which was used as a proxy for crowding.

**Household food security.** Household food security was assessed using the Household Food Insecurity Access Scale (HFIAS) [14]. Based on the total scores, households were classified into four categories: food secure (scores 0–2), moderately at risk (3–5), food insecure (6–10), and severely food insecure (11–27). For analyses, households were further dichotomised into food secure or insecure.

**Mental health measures.** Anxiety and depression were assessed using the Generalized Anxiety Disorder-7 (GAD-7), [15] and Patient Health Questionnaire-9 (PHQ-9) [16] respectively. Both self-administered tools are based on the Diagnostic and Statistical Manual of Mental Disorders criteria and provide validated measures of symptom severity and are not diagnostic therefore are considered probable anxiety or depression. Although developed in high-income settings, they have shown strong reliability and validity in Sowetan women [17] and other African populations [18]. The PHQ-9 includes 9 items rated from 0 ("Not at all") to 3 ("Nearly every day"), with scores categorised as minimal (0–4), mild (5–9), moderate (10–14), moderately severe (15–19), and severe (20–27). A score of ≥10 indicates probable depression. The GAD-7 includes 7 similarly scored items, categorising anxiety severity as minimal (0–4), mild (5–9), moderate (10–14), and severe (15–21), with probable anxiety defined as a score of ≥10. Adverse childhood experiences were measured using a 12-item ACE questionnaire, which documents experiences in three categories: emotional and/or physical abuse, sexual abuse, or household dysfunction. Based on the participants' retrospective report of specific adversities experienced over the first 18 years of their life, an overall ACE score was calculated using on the number of affirmative responses (yes response) where no = 0 and yes = 1. Three categories based on the level of childhood adversity exposure were calculated: scoring 0 (low exposure); 1–3 (intermediate exposure) and 4–12 (high exposure), as well as, clustered by quantity (low versus high) and type (household dysfunction versus abuse).

**Behavioural factors.** Alcohol use was assessed with the Alcohol Use Disorders Identification Test Consumption (AUDIT-C), a brief three-item screening tool evaluating the frequency and quantity of alcohol consumption, including binge drinking behaviour. Physical activity levels were determined through self-reported engagement in moderate or vigorous physical activity, using items adapted from the Global Physical Activity Questionnaire (GPAQ) [19]. Examples such as walking, running, sports, and labour-intensive work were provided to aid comprehension, and participants reported how often and for how long they engaged in these activities. Sleep quality was assessed using the Pittsburgh Sleep Quality Index (PSQI) [20], which evaluates components such as sleep duration, disturbances, latency, and overall quality over the past month; a global PSQI score greater than 5 was used to indicate poor sleep quality [21].

## Statistical analysis

Characteristics of participants were described using means and standard deviations and medians and interquartile ranges (IQRs) for continuous variables, and frequencies and proportions for categorical variables. We conducted stratification analyses to examine SES strata (lower vs higher) differences in socio-demographic, risk behaviour and outcome variables. For each outcome (probable anxiety; probable depression), we fitted hierarchical logistic regression models to quantify the incremental contribution of conceptually ordered determinants. Block 0 included age, education, relationship

status, parity, household SES (asset index), and household size. Block 1 added ACE (count). Block 2 added current food insecurity. Block 3 added health behaviours and sleep (physical activity hours/week, hazardous alcohol use, poor sleep quality). Block 4 added comorbid mental-health symptoms (moderate–severe anxiety for the depression model). At each step, we report adjusted odds ratios (95% CIs), changes in model fit (overall model significance, Δpseudo-R²), and likelihood-ratio tests comparing successive blocks. Women who exhibited both probable anxiety and depression were coded as a co-morbid and we also assessed factors associated with this group.

We specified a priori a multi-level mediation model grounded in social determinants and stress theories [22,23]. Socio-economic position (SEP) organises material and psychosocial exposures that become embodied across the life course and shape mental health. Building on the stress process, we posited that lower SEP increases chronic stressors and limits coping resources, elevating depressive symptoms via material (e.g., food insecurity), behavioural/bioregulatory (e.g., poor sleep, hazardous alcohol use) and social pathways (e.g., reduced social support). Collectively, these pathways operationalise the hypothesis that SEP associates with depression both directly and indirectly via modifiable mediators. Generalised Structural Equation Modelling (gSEM) was used to test for the mediation associations of the relationship between household SES and probable anxiety and depression. The model was assessed for goodness of fit using the Akaike Information Criterion (AIC) and Bayesian Information Criterion (BIC). Bidirectional pathways between mental health measures were formally tested and are presented in Model 1 and Model 2. Bidirectional paths between mental health measures and sleep were also evaluated and tested within the model; however, as the reverse path (from depression to sleep) was not statistically significant, the final model presented in the manuscript retained only the unidirectional path from sleep to mental health to improve model fit and interpretability in this study sample. This modelling decision does not discount the established bidirectional relationship between mental health and sleep; rather, it indicates that, in this sample, the unidirectional specification provided the best-fitting representation of the data. Direct, indirect, and total associations were calculated using non-linear combination estimates. All regression models were conducted using complete-case analysis, whereby participants with missing data on any variables included in each model were excluded from that analysis. To assess potential overlap between anxiety and depression symptoms, we conducted a confirmatory factor analysis (CFA) with latent variables for PHQ9 and GAD7.

## Results

Missing data was evident as some participants elected not to answer certain questions and varying sample sizes are reflected in the tables. Women who attended the data collection appointment had a mean age of 22.6 years and a median of 1 child per household. The majority of women had completed secondary schooling (57.8%), were in a committed relationship but not married (53.5%), and never been employed (59.4%) and these were similar across the SES groups (Table 1).

Behavioural and mental health risks of the women are outlined in Table 2 and Table 3 respectively. In this group of women, 20.0% reported consuming hazardous amounts of alcohol and 55.5% reported poor sleep. This was observed consistently across all SES groups. Women in the higher SES group reported greater levels of physical activity (high SES: 1.7 (SD 4.3) vs low SES: 1.4 (SD 3.8)), and 12.6% and 15.8% anxiety and depression respectively. 9.7% of the total sample of women experienced both depression and anxiety. The average ACE score reported by the women was 3.4 (SD 2.3), with 43.2% of women indicating high ACE exposure. A higher percentage of women in the lower SES group (47.4%) reported substantial childhood trauma compared to those in the higher SES group (40.3%).

We performed multivariable logistic regressions to further assess the odds of having either anxiety or depression (Table 4) in several models across various levels - household and individual (model 1), with childhood adversity (model 2), being food insecure (model 3), reporting behavioural (model 4) and a combined model (model 5) or experiencing both depression and anxiety concurrently (Table 5). We found that those who reported to be in a committed relationship but not married were less likely to report anxiety (OR 0.70 [95% CI 0.58; 0.84]) and depression (OR 0.66 [95% CI 0.56; 0.79]).

**Table 1. General household and individual characteristics of the young women participating in this study.**

| | Total population (n=7735) | Lower SES (n=2940) | Higher SES (n=4448) |
|---|---|---|---|
| **Household level** | | | |
| Household SES, sum of assets (mean (SD); median (IQR)); n=7388 | 8.0 (2.1) 8.0 (7 – 9) | 5.9 (1.3) 6 (5 – 7) | 9.3 (1.2) 9 (8 – 10) |
| Number of people living in household (median (IQR); max); n=7416 | 5 (4 - 7); 27 | 4 (3 – 6); 21 | 5 (4 – 7); 27 |
| Number of people per sleeping room (median (IQR); max); n=7401 | 2.5 (1.7 – 3.5); 15 | 2.5 (2 – 4); 15 | 2.3 (1.7 – 3.3); 15 |
| Household food insecurity (freq. Yes (%)); n=6551 | 2510 (38.3) | 1292 (51.6) | 1204 (30.1) |
| **Individual level** | | | |
| Age (years; mean (SD)); n=7700 | 22.6 (2.8) | 22.9 (2.8) | 22.4 (2.8) |
| Parity (median (IQR); max); n=7683 | 1 (0 – 1); 5 | 1 (0 – 1); 5 | 0 (0 – 1); 5 |
| Parity (freq. %); n=7683 | | | |
| 0 | 3646 (47.5) | 1166 (39.9) | 2332 (52.8) |
| 1 | 2969 (38.6) | 1212 (41.5) | 1605 (36.3) |
| 2 or more | 1068 (13.9) | 546 (18.6) | 481 (10.9) |
| Marital Status (freq. %); n=7421 | | | |
| Single | 3369 (45.4) | 1293 (44.1) | 2051 (46.3) |
| In a committed relationship but not married | 3969 (53.5) | 1609 (54.9) | 2329 (52.6) |
| Married | 83 (1.1) | 3 (1.0) | 51 (1.1) |
| Education (freq. (%)); n=7420 | | | |
| Some schooling but did not complete secondary education | 2688 (36.2) | 1343 (45.9) | 1324 (29.9) |
| Completed secondary education | 4284 (57.8) | 1466 (50.0) | 2782 (62.8) |
| Completed tertiary education | 448 (6.0) | 121 (4.1) | 325 (7.3) |
| Employment (freq. (%)); n=7435 | | | |
| Never employed | 4414 (59.4) | 1806 (61.5) | 2566 (57.8) |
| Not currently employed | 2415 (32.4) | 941 (32.1) | 1461 (32.9) |
| Currently employed | 606 (8.2) | 188 (6.4) | 413 (9.3) |
| Recipient of a Government grant (e.g., child support; yes; freq. (%)); n=7434 | 3646 (47.1) | 1617 (55.1) | 2003 (45.1) |

Abbreviations: n – number of participants; SES – socioeconomic status; SD – standard deviation; IQR – interquartile range. Total population includes all participants (n=7735); Lower SES and higher SES strata include only those with valid SES data (n=7388), thus the sum of these strata do not equal the total population.

**Table 2. Risk behaviours.**

| | Total population (n=7735) | Lower SES (n=2940) | Higher SES (n=4448) |
|---|---|---|---|
| Hazardous or harmful alcohol consumption (yes; freq. (%)); n=7704 | 1539 (20.0) | 587 (20.0) | 894 (20.1) |
| Moderate or vigorous physical activity (hours/week; mean (SD)); n=7700 | 1.6 (4.1) | 1.4 (3.8) | 1.7 (4.3) |
| Poor sleep quality (freq. Yes, (%)); n=7687 | 4264 (55.5) | 1608 (54.9) | 2473 (55.8) |

Abbreviations: n – number of participants; SES – socioeconomic status; SD – standard deviation.

Those reporting childhood adversity had 32–34% higher odds of reporting anxiety (ACE: OR 1.32 [95% CI 1.26; 1.37]; or depression (ACE: OR 1.34 [95% CI 1.30; 1.39]). Consuming a hazardous amount of alcohol increased the likelihood of mental health risk (depression: OR 1.29 [95% CI 1.03; 1.61]) while mental health risk increased more than three-fold in

**Table 3. Adverse childhood experiences, anxiety and depression.**

| | Total population (*n* = 7735) | Lower SES (*n* = 2940) | Higher SES (*n* = 4448) |
|---|---|---|---|
| **ACE** | | | |
| ACE (total count; mean (SD); median (IQR)); n = 7434 | 3.4 (2.3)<br>3 (2 − 5) | 3.7 (2.3)<br>3 (2 − 5) | 3.2 (2.2)<br>3 (2 − 5) |
| ACE (freq. Yes (%)) 4 or more | 3208 (43.2) | 1391 (47.4) | 1788 (40.3) |
| **Anxiety** | | | |
| Moderate-severe anxiety (freq. Yes (%)); n = 7702 | 971 (12.6) | 371 (12.7) | 562 (12.7) |
| **Depression** | | | |
| Depression (freq. (yes; %) based upon PHQ-9 or on medication); n = 7702 | 1217 (15.8) | 471 (16.0) | 694 (15.6) |
| **Comorbidity** | | | |
| Depression (freq. (yes; %) based upon PHQ-9 or on medication) and moderate-severe anxiety (freq. Yes (%); n = 7702 | 747 (9.7) | 288 (9.8) | 426 (9.6) |

Abbreviations: *n* – number of participants; SES – socioeconomic status; SD – standard deviation; IQR – interquartile range; ACE – adverse childhood experiences; PHQ9 – Patient Health Questionnaire 9.

those who reported poor sleep (anxiety: OR 4.03 [95% CI 3.07; 5.29]; depression: OR 3.18 [95% CI 2.54; 3.99]). Additionally, those who reported moderate to severe level of anxiety were more than 30 times more likely also to report depressive symptoms (OR 32.20 [95% CI 24.34; 42.60]). Overall model significance, goodness of fit, and pseudo R² were evaluated at each step of model building. From Model 1 through to Model 5 of the logistic regressions, model χ² and degrees of freedom increased, indicating progressive improvement in model fit, while the −2 Log Likelihood values decreased correspondingly. The final model demonstrated the best fit, with a pseudo R² of 0.50, suggesting that approximately half of the variance in the outcome was explained by the included factors. CFA supported distinct latent factors for anxiety and depression. In model 5, removal of the overlapping GAD7 item attenuated the OR for anxiety predicting depression, however, the association remained highly significant (OR 26.9 [95% CI 20.1; 36.0]) (results not shown), indicating that the strong association was not fully attributable to symptom overlap.

In the model assessing comorbid depression and anxiety (**Table 5**), results mirrored those found for each condition individually. Individuals in committed relationships had lower odds of reporting both conditions concurrently (OR 0.67 [95% CI 0.54; 0.84]), while higher parity was associated with 29% higher odds of comorbid anxiety and depression (OR 1.29 [95% CI 1.08; 1.52]). Those with a history of childhood adversity were 31% more likely to report comorbidity (OR 1.31 [95% CI 1.25; 1.38]). Poor sleep was strongly associated with comorbid depression and anxiety, with more than fourfold increased odds (OR 4.03 [95% CI 3.07; 5.29]).

A gSEM was constructed *a priori* to assess the impact of household and individual level variables on mental health risk (**Fig 1** and **Table 6**). We assessed the bidirectional relationship between depression and anxiety using two models: Model 1 with depression as the outcome and anxiety as the mediator (AIC: 72603.36; BIC: 72914.14), and Model 2 with anxiety as the outcome and depression as the mediator (AIC: 72602.00; BIC: 72912.77). The results revealed significant direct associations of household SES on alcohol (*p* = 0.015); childhood adversity on anxiety and depression (*p* < 0.001); and marital status on anxiety (*p* < 0.001). The results revealed several important mediations. Firstly, household poverty associated with anxiety partially via sleep (54.2%) and fully via depression (86.9%), while its association on depression was fully mediated by sleep (43.7%). Childhood adversity was partially mediated by its association on depression through anxiety (79.4%), sleep disturbances (31.3%), and alcohol use (14.2%), and on anxiety via depression (88.6%) and sleep (42.2%). Marital status impacted anxiety partially through depression (81.4%), and depression was also partially mediated by anxiety (85.5%). Additionally, alcohol use associated with anxiety with full mediation through depression (96.5%). Notably,

**Table 4. Determining factors associated with anxiety and depression (logistic regressions).**

|  |  | Anxiety; yes | | Depression; yes | |
|---|---|---|---|---|---|
|  |  | OR | (95% CI) | OR | (95% CI) |
| **Model 1** (n=4145) | Age (years) | 0.98 | (0.95; 1.02) | 0.98 | (0.95; 1.02) |
|  | Completed secondary education (yes); reference did not complete secondary education | 1.11 | (0.91; 1.35) | 1.06 | (0.89; 1.27) |
|  | Completed tertiary education (yes); reference did not complete secondary education | 0.92 | (0.59; 1.43) | 1.07 | (0.73; 1.56) |
|  | In a committed relationship (yes) | **0.70***** | (0.58; 0.84) | **0.66***** | (0.56; 0.79) |
|  | Parity | **1.27**** | (1.10; 1.47) | 1.14 | (0.99; 1.30) |
|  | Household SES (sum of assets) | 0.99 | (0.95; 1.04) | 0.99 | (0.95; 1.04) |
|  | Number of people living in household | 0.98 | (0.93; 1.04) | 0.97 | (0.92; 1.02) |
|  | Overall model significance (Chi square; df; p value) | 25.86; 7; <0.001 | | 27.87; 7; <0.001 | |
|  | Goodness of fit (-2 Log Likelihood) | 3045.58 | | 3535.01 | |
|  | Pseudo $R^2$ (Nagekerke $R^2$) | 0.01 | | 0.01 | |
| **Model 2** (n=4144) | Age (years) | 0.98 | (0.94; 1.02) | 0.98 | (0.94; 1.01) |
|  | Completed secondary education (yes); reference did not complete secondary education | 1.15 | (0.94; 1.41) | 1.11 | (0.92; 1.33) |
|  | Completed tertiary education (yes); reference did not complete secondary education | 1.01 | (0.64; 1.60) | 1.20 | (0.81; 1.79) |
|  | In a committed relationship (yes) | **0.69***** | (0.57; 0.83) | **0.64***** | (0.54; 0.77) |
|  | Parity | **1.23**** | (1.06; 1.43) | 1.10 | (0.95; 1.26) |
|  | Household SES (sum of assets) | 1.02 | (0.97; 1.07) | 1.02 | (0.98; 1.07) |
|  | Number of people living in household | 0.96 | (0.91; 1.02) | **0.95*** | (0.90; 1.00) |
|  | ACE (total count) | **1.32***** | (1.26; 1.37) | **1.34***** | (1.30; 1.39) |
|  | Overall model significance (Chi square; df; p value) | 213.09; 8; <0.001 | | 284. 27; 8; <0.001 | |
|  | Goodness of fit (-2 Log Likelihood) | 2858.10 | | 3278.27 | |
|  | Pseudo $R^2$ (Nagekerke $R^2$) | 0.10 | | 0.12 | |
| **Model 3** (n=3604) | Age (years) | 0.97 | (0.93; 1.01) | 0.97 | (0.93; 1.01) |
|  | Completed secondary education (yes); reference did not complete secondary education | 1.10 | (0.88; 1.38) | 1.04 | (0.85; 1.27) |
|  | Completed tertiary education (yes); reference did not complete secondary education | 0.76 | (0.43; 1.35) | 1.05 | (0.66; 1.66) |
|  | In a committed relationship (yes) | **0.67***** | (0.54; 0.82) | **0.65***** | (0.54; 0.79) |
|  | Parity | **1.26**** | (1.07; 1.49) | 1.05 | (0.90; 1.23) |
|  | Household SES (sum of assets) | 1.03 | (0.97; 1.08) | 1.03 | (0.98; 1.09) |
|  | Number of people living in household | 0.96 | (0.91; 1.02) | **0.94*** | (0.89; 0.99) |
|  | ACE (total count) | **1.36***** | (1.30; 1.42) | **1.40***** | (1.34; 1.46) |
|  | Food insecurity (yes) | 1.10 | (0.85; 1.41) | 0.97 | (0.77; 1.21) |
|  | Overall model significance (Chi square; df; p value) | 227.59; 9; <0.001 | | 298.42; 9; <0.001 | |
|  | Goodness of fit (-2 Log Likelihood) | 2366.82 | | 2763.36 | |
|  | Pseudo $R^2$ (Nagekerke $R^2$) | 0.12 | | 0.14 | |
| **Model 4** (n=3594) | Age (years) | 0.96 | (0.92; 1.01) | 0.96 | (0.92; 1.00) |
|  | Completed secondary education (yes); reference did not complete secondary education | 1.06 | (0.84; 1.33) | 1.00 | (0.82; 1.24) |
|  | Completed tertiary education (yes); reference did not complete secondary education | 0.70 | (0.39; 1.26) | 1.02 | (0.63; 1.62) |
|  | In a committed relationship (yes) | **0.67***** | (0.54; 0.84) | **0.66***** | (0.54; 0.80) |
|  | Parity | **1.29***** | (1.08; 1.52) | 1.06 | (0.90; 1.24) |
|  | Household SES (sum of assets) | 1.02 | (0.97; 1.08) | 1.03 | (0.98; 1.08) |
|  | Number of people living in household | 0.96 | (0.90; 1.02) | **0.93*** | (0.88; 0.99) |
|  | ACE (total count) | **1.32***** | (1.25; 1.38) | **1.35***** | (1.30; 1.41) |
|  | Food insecurity (yes) | 1.05 | (0.81; 1.36) | 0.93 | (0.74; 1.16) |
|  | Moderate or vigorous physical activity (hours/week) | 1.01 | (0.98; 1.04) | 0.99 | (0.97; 1.02) |

*(Continued)*

**Table 4.** (Continued)

| | | Anxiety; yes | | Depression; yes | |
|---|---|---|---|---|---|
| | | OR | (95% CI) | OR | (95% CI) |
| | Hazardous alcohol consumption (yes) | 1.09 | (0.85; 1.39) | **1.29*** | (1.03; 1.61) |
| | Poor sleep quality (yes) | **4.03*** | (3.07; 5.29) | **3.18*** | (2.54; 3.99) |
| | Overall model significance (Chi square; df; p value) | 352.69; 12; <0.001 | | 417.96; 12; <0.001 | |
| | Goodness of fit (-2 Log Likelihood) | 2235.18 | | 2633.65 | |
| | Pseudo R$^2$ (Nagekerke R$^2$) | 0.18 | | 0.19 | |
| **Model 5** (n = 3594) | Age (years) | – | – | 0.97 | (0.92; 1.02) |
| | Completed secondary education (yes); reference did not complete secondary education | – | – | 0.97 | (0.75; 1.26) |
| | Completed tertiary education (yes); reference did not complete secondary education | – | – | 1.27 | (0.73; 2.22) |
| | In a committed relationship (yes) | – | – | **0.73*** | (0.57; 0.93) |
| | Parity | – | – | 0.88 | (0.72; 1.08) |
| | Household SES (sum of assets) | – | – | 1.03 | (0.97; 1.09) |
| | Number of people living in household | – | – | **0.93*** | (0.86; 1.00) |
| | ACE (total count) | – | – | **1.28*** | (1.21; 1.35) |
| | Food insecurity (yes) | – | – | 0.88 | (0.67; 1.16) |
| | Moderate or vigorous physical activity (hours/week) | – | – | 0.98 | (0.95; 1.01) |
| | Hazardous alcohol consumption (yes) | – | – | **1.40*** | (1.07; 1.84) |
| | Poor sleep quality (yes) | – | – | **2.05*** | (1.57; 2.68) |
| | Anxiety (moderate-severe) | – | – | **32.20*** | (24.34; 42.60) |
| | Overall model significance (Chi square; df; p value) | – | | 1145.82; 13; <0.001 | |
| | Goodness of fit (-2 Log Likelihood) | – | | 1905.78 | |
| | Pseudo R$^2$ (Nagekerke R$^2$) | – | | 0.48 | |

Abbreviations: SES: socioeconomic status. Bold values denote statistical significance *$p < 0.05$; **$p < 0.01$; ***$p < 0.001$.

anxiety was found to fully mediate the relationship between depression and severe food insecurity, accounting for 92.3% of the total association. Bidirectional relationships between mental health and sleep were also assessed as seen in Model 3 (S1 Table) (AIC: 72617.2; BIC: 72927.98); household SES remained directly associated with sleep ($p = 0.005$), with anxiety partially mediating 56.8% of the total association, while depression showed no mediation.

## Discussion

Our findings revealed a high prevalence of young women with reported anxiety and depressive symptoms, and interconnected links between structural and individual risk factors. The reported anxiety prevalence was similar to that of the general population provincial average (15%) and national average (17%) in South Africa in 2022, but three times more than the WHO estimated global average (4%) [9]. In contrast, the reported depression prevalence was less than the general population provincial average (24%) and national average (26.2%), but still three times more than the estimated global average (3.8%). Importantly, 1 in 10 women reported both anxiety and depression symptoms.

Nearly half of the women surveyed reported high ACEs, a rate almost three times greater than the national prevalence of 14.6% [9]. This contrast underscores the significantly heightened prevalence of childhood adversity among young women residing in Soweto. Numerous studies confirm the strong relationship between ACEs and poor adult mental health [9,24]. Adverse experiences over the first 18 years of life, including physical, sexual, psychological abuse and neglect; parental loss due to death, divorce, or separation; parental mental illness or substance use; and poverty, have been consistently linked to depression and anxiety in South African adults [9]. Gendered disparities may further compound these

**Table 5. Identifying factors associated with anxiety and depression co-morbidity (logistic regression) (*n* = 3954).**

| | Anxiety and depression co-morbidity (yes) | |
| --- | --- | --- |
| | OR | (95% CI) |
| Age (years) | 0.96 | (0.92; 1.01) |
| Completed secondary education (yes); reference did not complete secondary education | 1.06 | (0.84; 1.33) |
| Completed tertiary education (yes); reference did not complete secondary education | 0.70 | (0.39; 1.26) |
| In a committed relationship (yes) | **0.67***** | (0.54; 0.84) |
| Parity | **1.29***** | (1.08; 1.52) |
| Household SES (sum of assets) | 1.02 | (0.97; 1.08) |
| Number of people living in household | 0.96 | (0.90; 1.02) |
| ACE (total count) | **1.31***** | (1.25; 1.38) |
| Food insecurity (yes) | 1.05 | (0.81; 1.36) |
| Moderate or vigorous physical activity (hours/week) | 1.01 | (0.98; 1.04) |
| Hazardous alcohol consumption (yes) | 1.09 | (0.85; 1.39) |
| Poor sleep quality (yes) | **4.03***** | (3.07; 5.29) |

Abbreviations: SES: socioeconomic status. Bold values denote statistical significance *$p < 0.05$; **$p < 0.01$; ***$p < 0.001$.

risks, as women have been shown to be more likely to experience harsh early living conditions, including food insecurity, emotional neglect, and sexual abuse [25]. These long-term psychological effects are reflected in the elevated levels of mental health symptoms observed in this cohort, particularly among those with high ACE exposure, supporting the evidence that early life adversity plays a central role in shaping adult mental health outcomes. The data further indicated a pronounced disparity linked to SES, with a greater proportion of women from lower household SES groups reporting greater ACE exposure compared to their peers in higher SES categories. The elevated rates of childhood adversity observed may, in part, reflect the compounded and pervasive stressors associated with urban poverty. These include exposure to violence, unstable family environments, and restricted access to robust social support systems [26,27].

Risk behaviours were prevalent among this cohort of young women, with hazardous alcohol consumption and poor sleep quality emerging as particularly significant concerns. There are robust association data between hazardous alcohol use, poor sleep quality, and an elevated risk of mental health disorders [28,29]. Moreover, hazardous alcohol consumption may compound risk, functioning dually as a maladaptive coping strategy and a catalyst for heightened psychological distress, potentially exacerbating symptoms of anxiety and depression. The considerable prevalence of poor sleep within this group of women may be indicative of chronic stress, however within this specific context, it may be more likely due to environmental conditions, such as household overcrowding, noise and disrupted sleep patterns.

For the women in this study, being in a committed relationship was protective, as these women were less likely to report symptoms of anxiety and depression – possibly reflecting emotional and financial support provided by stable partnerships. This is consistent with our previous qualitative research findings, which highlighted the critical value of relational and social support for young women living in Soweto [30]. In contrast, exposure to childhood trauma was significantly associated with poorer mental health outcomes. Women with a history of ACEs were over 30% more likely, more likely to experience anxiety or depression symptoms. These findings underscore the enduring psychological effects of early life trauma within contexts of economic hardship. This aligns with previous South African research showing that the odds of probable depression or anxiety increased by more than 1.2 times for each standard deviation increase in the ACE score

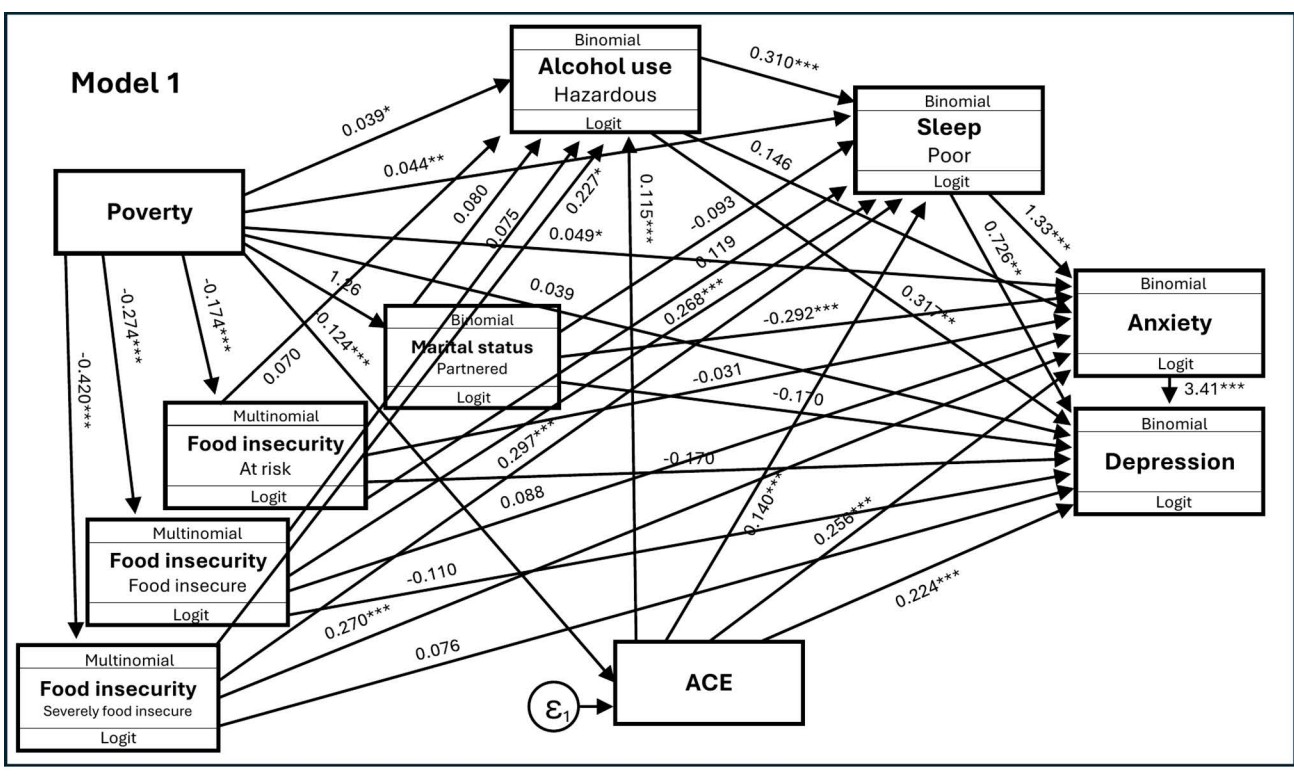

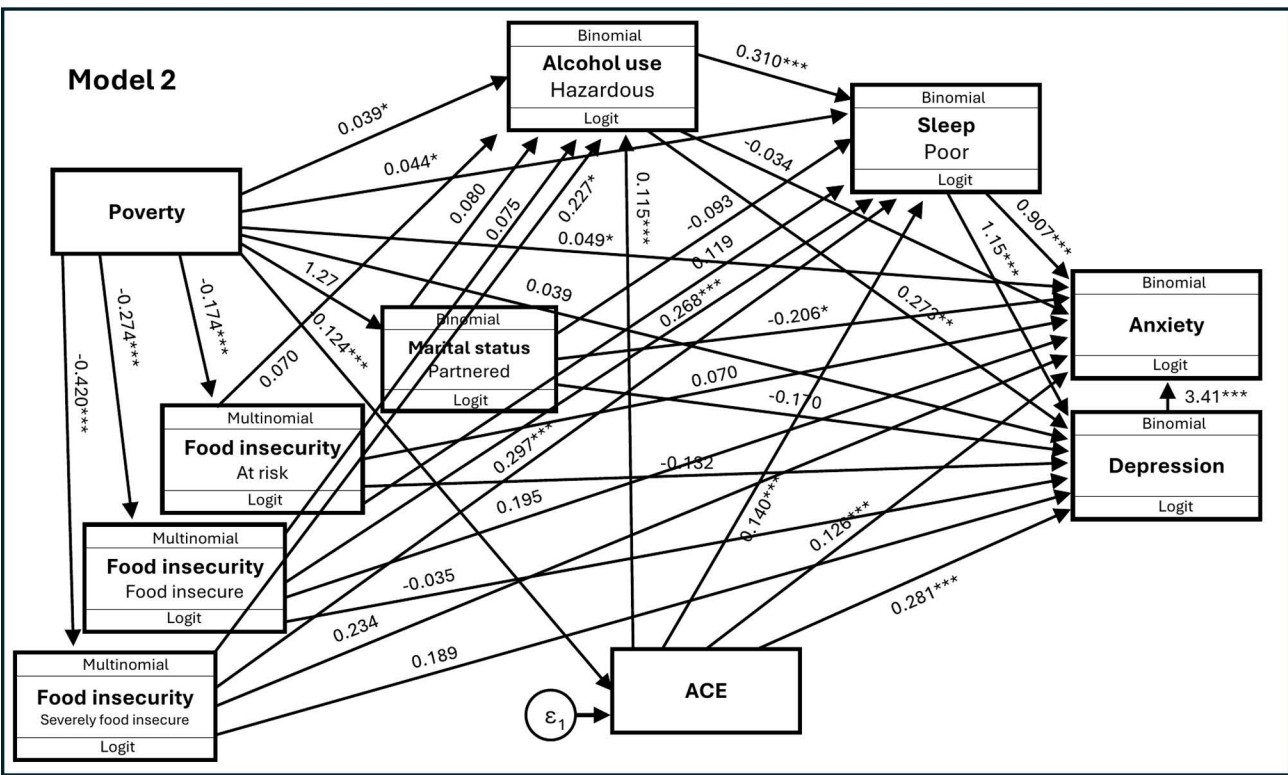

**Fig 1. Generalised structural equation model for sociodemographic and mental health, showing model 1 (depression as outcome, anxiety as mediator) and model 2 (anxiety as outcome, depression as mediator).** Abbreviations: SES – socioeconomic status; ACE – adverse childhood experiences. * p < 0.05; ** p ≤ 0.001.

**Table 6. Generalised structural equation model in a sample of respondents for socioeconomic status, marital status, sleep and mental health (*n* = 7377).**

| Exposure | Outcome (*n* = 6467) | Direct β (CI) | *p* value | Indirect β (CI) | *p* value | Total β (CI) | *p* value | % mediated |
|---|---|---|---|---|---|---|---|---|
| **Model 1** | | | | | | | | |
| **Poverty on mental health via ACE (score)** | | | | | | | | |
| Poverty | GAD7 > 10 (probable) | 0.048 (0.007; 0.090) | **0.021** | -0.032 (-0.039; -0.024) | **<0.001** | 0.017 (-0.025; 0.058) | 0.43 | ‡ |
| Poverty | PHQ9 > 10 (probable) | 0.039 (-0.007; 0.085) | 0.093 | -0.028 (-0.035; -0.020) | **<0.001** | 0.011 (-0.034; 0.057) | 0.63 | ‡ |
| **Poverty on depression via anxiety (binary)** | | | | | | | | |
| Poverty | PHQ9 > 10 (probable) | 0.039 (-0.007; 0.085) | 0.093 | 0.165 (0.024; 0.306) | **0.021** | 0.204 (0.057; 0.352) | **0.007** | 80.9%* |
| **ACE on depression via anxiety (binary)** | | | | | | | | |
| ACE | PHQ9 > 10 (probable) | 0.224 (0.183; 0.264) | **<0.001** | 0.873 (0.742; 1.00) | **<0.001** | 1.10 (0.960; 1.23) | **<0.001** | 79.4%† |
| **Marital status on depression via anxiety (binary)** | | | | | | | | |
| Partnered | PHQ9 > 10 (probable) | -0.170 (-0.347; 0.007) | 0.060 | -1.00 (-1.54; -0.457) | **<0.001** | -1.17 (-1.73; -0.599) | **<0.001** | 85.5%* |
| **Poverty on mental health via alcohol use (binary)** | | | | | | | | |
| Poverty | PHQ9 > 10 (probable) | 0.039 (-0.007; 0.085) | 0.093 | 0.012 (-0.000; 0.025) | 0.056 | 0.051 (0.004; 0.098) | **0.033** | – |
| Poverty | GAD7 > 10 (probable) | 0.048 (0.007; 0.090) | **0.021** | 0.006 (-0.003; 0.014) | 0.19 | 0.054 (0.012; 0.096) | **0.011** | – |
| **Poverty on mental health via poor sleep (binary)** | | | | | | | | |
| Poverty | PHQ9 > 10 (probable) | 0.039 (-0.007; 0.085) | 0.093 | 0.031 (0.011; 0.052) | **0.002** | 0.071 (0.021; 0.120) | **0.005** | 43.7%* |
| Poverty | GAD7 > 10 (probable) | 0.048 (0.007; 0.090) | **0.021** | 0.058 (0.023; 0.093) | **0.001** | 0.107 (0.053; 0.160) | **<0.001** | 54.2%† |
| **Poverty on mental health via food insecurity (at risk)** | | | | | | | | |
| Poverty | PHQ9 > 10 (probable) | 0.039 (-0.007; 0.085) | 0.093 | 0.029 (-0.014; 0.074) | 0.18 | 0.069 (0.010; 0.128) | **0.022** | – |
| Poverty | GAD7 > 10 (probable) | 0.048 (0.007; 0.090) | **0.021** | 0.005 (-0.034; 0.045) | 0.79 | 0.054 (0.000; 0.107) | **0.048** | – |
| **Poverty on mental health via food insecurity (food insecure)** | | | | | | | | |
| Poverty | PHQ9 > 10 (probable) | 0.039 (-0.007; 0.085) | 0.093 | 0.030 (-0.039; 0.099) | 0.39 | 0.069 (-0.007; 0.145) | 0.073 | – |
| Poverty | GAD7 > 10 (probable) | 0.048 (0.007; 0.090) | **0.021** | -0.024 (-0.086; 0.038) | 0.44 | 0.024 (-0.043; 0.092) | 0.48 | – |
| **Poverty on mental health via food insecurity (severely food insecure)** | | | | | | | | |
| Poverty | PHQ9 > 10 (probable) | 0.039 (-0.007; 0.085) | 0.093 | -0.032 (-0.141; 0.077) | 0.57 | 0.007 (-0.098; 0.112) | 0.89 | – |
| Poverty | GAD7 > 10 (probable) | 0.048 (0.007; 0.090) | **0.021** | -0.114 (-0.210; -0.017) | **0.021** | -0.065 (-0.158; 0.028) | 0.17 | ‡ |
| **ACE on mental health via poor sleep** | | | | | | | | |
| ACE | PHQ9 > 10 (probable) | 0.224 (0.183; 0.264) | **<0.001** | 0.102 (0.069; 0.135) | **<0.001** | 0.326 (0.275; 0.376) | **<0.001** | 31.3%† |
| ACE | GAD7 > 10 (probable) | 0.256 (0.221; 0.291) | **<0.001** | 0.187 (0.144; 0.229) | **<0.001** | 0.443 (0.389; 0.497) | **<0.001** | 42.2%† |
| **ACE on mental health via alcohol use (binary)** | | | | | | | | |
| ACE | PHQ9 > 10 (probable) | 0.224 (0.183; 0.264) | **<0.001** | 0.037 (0.012; 0.061) | **0.004** | 0.260 (0.214; 0.306) | **<0.001** | 14.2%† |
| ACE | GAD7 > 10 (probable) | 0.256 (0.221; 0.291) | **<0.001** | 0.017 (-0.004; 0.038) | 0.12 | 0.273 (0.233; 0.313) | **<0.001** | – |
| **Alcohol use on depression via anxiety (binary)** | | | | | | | | |
| Hazardous alcohol | PHQ9 > 10 (probable) | 0.317 (0.115; 0.519) | **0.002** | 0.498 (-0.121; 1.12) | 0.12 | 0.815 (0.163; 1.47) | **0.014** | – |
| **Food insecurity on depression via anxiety (binary)** | | | | | | | | |
| At risk | PHQ9 > 10 (probable) | -0.170 (-0.419; 0.079) | 0.18 | -0.104 (-0.876; 0.667) | 0.79 | -0.274 (-1.09; 0.536) | 0.51 | – |
| Food insecure | PHQ9 > 10 (probable) | -0.110 (-0.361; 0.142) | 0.39 | 0.299 (-0.466; 1.06) | 0.44 | 0.189 (-0.616; 0.995) | 0.65 | – |
| Severely food insecure | PHQ9 > 10 (probable) | -0.076 (-0.183; 0.335) | 0.57 | 0.921 (0.139; 1.70) | **0.021** | 0.998 (0.174; 1.82) | **0.018** | 92.3%* |
| **Model 2** | | | | | | | | |
| **Poverty on anxiety via depression (binary)** | | | | | | | | |
| Poverty | GAD7 > 10 (probable) | 0.026 (-0.025; 0.077) | 0.32 | 0.173 (0.045; 0.301) | **0.008** | 0.199 (0.061; 0.337) | **0.005** | 86.9%* |
| **ACE on anxiety via depression (binary)** | | | | | | | | |
| ACE | GAD7 > 10 (probable) | 0.126 (0.081; 0.171) | **<0.001** | 0.957 (0.832; 1.08) | **<0.001** | 1.08 (0.953; 1.21) | **<0.001** | 88.6%† |

*(Continued)*

**Table 6.** (Continued)

| Exposure | Outcome (*n* = 6467) | Direct β (CI) | *p* value | Indirect β (CI) | *p* value | Total β (CI) | *p* value | % mediated |
|---|---|---|---|---|---|---|---|---|
| **Marital status on anxiety via depression (binary)** | | | | | | | | |
| Partnered | PHQ9 > 10 (probable) | -0.206 (-0.401; -0.011) | **0.038** | -0.904 (-1.40; -0.412) | **<0.001** | -1.11 (-1.64; -0.581) | **<0.001** | 81.4%† |
| **Alcohol use on anxiety via depression (binary)** | | | | | | | | |
| Hazardous alcohol | GAD7 > 10 (probable) | -0.034 (-0.261; 0.193) | 0.77 | 0.931 (-0.370; 1.49) | **0.001** | 0.965 (0.292; 1.50) | **0.004** | 96.5%* |
| **Food insecurity on anxiety via depression (binary)** | | | | | | | | |
| At risk | GAD7 > 10 (probable) | 0.070 (-0.207; 0.348) | 0.62 | -0.450 (-1.15; 0.247) | 0.21 | -0.380 (-1.13; 0.375) | 0.32 | – |
| Food insecure | GAD7 > 10 (probable) | 0.195 (-0.081; 0.471) | 0.17 | -0.118 (-0.816; 0.579) | 0.74 | 0.077 (-0.673; 0.827) | 0.84 | – |
| Severely food insecure | GAD7 > 10 (probable) | 0.234 (-0.051; 0.519) | 0.11 | 0.645 (-0.066; 1.36) | 0.075 | 0.880 (0.113; 1.65) | **0.024** | – |

Analytic sample size corresponds to the outcome of interest; sample size varied across paths due to missing data. Abbreviations: SES – socioeconomic status; PHQ9 – patient health questionnaire 9; GAD7 – generalised anxiety disorder 7; ACE – adverse childhood experience; FIS – food insecurity. * Full mediation, *p* < 0.05; †partial mediation, *p* < 0.05; ‡ inconsistent mediation, *p* < 0.05.

[9,31]. Moreover, hazardous alcohol consumption was associated with modestly increased odds of anxiety and substantially higher odds for depression. Women who reported moderate to severe anxiety were also 32 times more likely to experience depressive symptoms. The coexistence of depression and anxiety has been recognised, with documented findings showing that anxiety disorders are frequently comorbid with major depressive disorder, and that 50–60% of individuals with major depressive disorder report a lifetime history of one or more anxiety disorders [32].

To better understand the interplay between these factors, a gSEM developed *a priori* and informed by the regression results, underscored the complex relationship between poverty (household SES), psychosocial stressors (i.e., childhood adversity, food insecurity), behavioural risk (i.e., alcohol use, poor sleep), and mental health outcomes among this cohort of young Sowetan women. Significant direct associations were observed for household SES on alcohol use, childhood adversity on both anxiety and depression, and marital status on anxiety, indicating that structural and relationship factors continue to shape mental health risk. Importantly, several key mediational pathways emerged. Childhood adversity associated with both depression and anxiety indirectly – largely through anxiety, sleep disturbances, and alcohol use for depression, and through depression and sleep for anxiety. Household SES affected anxiety partially through sleep and fully through depression, and its impact on depression was fully mediated by sleep. Furthermore, committed relationship status shaped anxiety partly through depression and was also fully associated with depression via anxiety, potentially suggesting that economic hardships may, in part, contribute to poor mental health via disruptions in relationship stability (i.e., committed relationship), emotional wellbeing (i.e., depression), and rest (i.e., sleep), corroborated by earlier evidence [33,34]. Notable, anxiety fully mediated the association between depression and severe food insecurity, highlighting anxiety as a central pathway linking emotional distress and material deprivation in this sample. This reinforces the interdependence of relational, emotional and structural factors in shaping mental health outcomes [35]. Importantly, the association of alcohol use on anxiety was fully mediated through depression, indicating that alcohol-related mental health risks may be predominantly bi-directionally driven by underlying depressive symptoms [36]. This model empirically supports theoretical frameworks on the social determinants of mental health in low-resource urban settings and quantifies how early-life and current stressors interact to shape mental health outcomes.

These findings draw attention to the ways in which stressors associated with systemic barriers to well-being within contexts such as Soweto are persisting in South Africa. Systems, such as apartheid spatial planning and overcrowded housing conditions, continue to place people living in these contexts at a socioeconomic disadvantage [37], thus contributing to poor mental health. The term "structural violence", defined as the social structures that includes economic

and political systems that prevent individuals, families, communities and societies from meeting their full potential [38]. Applying this concept, it is then easy to comprehend that the structural violence risk that young women in contexts such as Soweto have to contend with negatively impacts their mental health, and that living with this level of risk may contribute to their poor sleep quality. Also, for women of reproductive age, periods such as pregnancy and the postpartum phase are particularly sensitive, with maternal mental health having documented effects on both mothers and their children [39]. While not all women will become mothers, for those who may, early identification and support for mental health conditions in the preconception period, may offer a critical opportunity to interrupt cycles of adversity and improve outcomes for both women and their offspring.

There are several limitations of this study worth noting. Firstly, while we document strong associations with early-life trauma, we neither directly measured nor included structural exposures, nor did we link participants to area-level administrative data on crime, or service density; future research could integrate geocoded deprivation indices to model structural pathways. SES was assessed using a household asset score, which, although commonly applied, may not fully differentiate between the heterogeneity within a similar urban context, potentially attenuating observed associations related to social structure. Our cross-sectional design further limits causal inference, including conclusions regarding the directionality of associations (i.e., sleep and depression) and bidirectional relationships cannot be excluded. Lastly, women self-reported symptoms in a face-to-face interview, which may have introduced social desirability bias. While appropriate for population studies, future approaches using objective measures may help reduce misreporting and improve mental health assessment accuracy, as highlighted in recent work on EEG-based monitoring methods [40].

In conclusion, we show that socioeconomic stress compounded by early-life trauma co-occur and are associated with elevated anxiety and depression among young women in urban poor settings. These risks are not individual failings but the likely consequence of structural injustice. Effective responses should pair trauma-informed clinical care with upstream action on structural violence and childhood adversity, while amplifying the intrapersonal strengths and social support that many young women already mobilise. A practical path could be to leverage Youth Safe Hubs as integrated, one-stop access points delivering: (i) routine mental-health screening; (ii) a brief, evidence-based problem-solving session; (iii) same-day referral to a co-located social worker for grant eligibility and protection services; (iv) short-term food support where indicated; and (v) linkage to skills training and apprenticeships to improve employability. Embedding this bundle within youth-centred, rights-based, and community-led platforms creates a feasible, scalable model that addresses both proximal symptoms and distal drivers. Future work should evaluate implementation fidelity, cost-effectiveness, and mental-health, social, and economic outcomes, ensuring that evidence generated directly informs policy and investment to reduce inequities and improve the life chances of young women in South Africa.

## Supporting information

**S1 Table. Generalised structural equation model in a sample of respondents for socioeconomic status, marital status, sleep and mental health (n = 7377).**
(DOCX)

**S1 Fig. Consort Diagram.**
(TIFF)

## Author contributions

**Conceptualization:** Shane Norris, Stephen Lye.

**Data curation:** Wihan Taljaard.

**Formal analysis:** Lukhanyo H. Nyati, Ashleigh Craig.

**Methodology:** Shane Norris.

**Supervision:** Shane Norris.

**Writing – original draft:** Shane Norris, Claire Hart, Ashleigh Craig.

**Writing – review & editing:** Shane Norris, Claire Hart, Lukhanyo H. Nyati, Wihan Taljaard, Rayjean Hung, Ravi Retnakaran, Stephen Lye, Catherine Draper, Ashleigh Craig.

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
