## [Decision Letter · Decision Letter 0]

7 Oct 2025

PMEN-D-25-00305

Preconception mental health (Healthy Life Trajectories Initiative): Identifying factors associated with probable anxiety and depression among young women living in urban-poor South Africa

PLOS Mental Health

Dear Dr. Norris,

Thank you for submitting your manuscript to PLOS Mental Health. After careful consideration, we feel that it has merit but does not fully meet PLOS Mental Health’s publication criteria as it currently stands. Therefore, we invite you to submit a revised version of the manuscript that addresses the points raised during the review process.

We look forward to receiving your revised manuscript.

Kind regards,

Gellan Karamallah Ramadan Ahmed

Academic Editor

PLOS Mental Health

Journal Requirements:

Additional Editor Comments (if provided):

Reviewers' comments:

Reviewer's Responses to Questions

**Comments to the Author**

1. Does this manuscript meet PLOS Mental Health’s publication criteria?

Reviewer #1: Partly

Reviewer #2: No

2. Has the statistical analysis been performed appropriately and rigorously?

Reviewer #1: Yes

Reviewer #2: No

3. Have the authors made all data underlying the findings in their manuscript fully available (please refer to the Data Availability Statement at the start of the manuscript PDF file)?

Reviewer #1: Yes

Reviewer #2: No

4. Is the manuscript presented in an intelligible fashion and written in standard English?

Reviewer #1: Yes

Reviewer #2: Yes

Reviewer #1: This is an important study aimed at unraveling the factors associated with increased risk of mental health conditions in vulnerable populations with the purpose of establishing mental health support strategies. The authors stated that their goal is to investigate how socioeconomic status affects anxiety and depression in urban Africa, while also referencing the well-established link between poverty and common mental health conditions.

They further detailed the key mediators through which the socioeconomic status framework can be measured. In this twofold study, the authors employed a cross-sectional study design with vast data collection and a wide array of location representation. They clearly stated the factors that disqualified subjects from participating in this study and the number of informed consents obtained for the study. The time period of data collection was stated, and the source of ethical approval was disclosed. It was encouraging to note that they took the positive steps of referring participants identified as needing mental health services to the local health services. The data collection technique was clearly disclosed, with a household asset score computed considering the various demographic information disclosed by participants. This score, used as the economic differentiation tool for assessing mental health measures and behavioral factors, was also disclosed

Statistical analysis of the results obtained and the coding method were extensively tabulated. However, the study's purpose remains somewhat unclear without suggesting a definite path towards addressing apartheid influenced systemic barriers, which contribute to the increase in anxiety and depression for the socioeconomically disadvantaged young women population in South Africa. Since the study conclusion is substantiated with evidence that early life trauma is a precursor to the increased risk for anxiety and depression among young women in urban poor settings, a glaring study limitation still exists because the study results only loosely mentioned the need for mental health, social support for the vulnerable population. It fails to identify concrete steps for addressing structural inequalities and social injustices, as these unfortunately play a significant role in creating the environment upon which anxiety/depression breeds in later years in the lives of young women living in Urban poor South Africa.

Reviewer #2: My negative response to Question 1 is primarily because the manuscript, in its current form, does not fully meet PLOS Mental Health’s publication Criterion #4: "Conclusions are presented in an appropriate fashion and are supported by the data." The conclusions, particularly those derived from the generalized structural equation modeling (gSEM), are currently overstated.

For instance, the interpretation of the generalized structural equation model (gSEM) presents a simplified, unidirectional view of what are likely complex, bidirectional relationships. The modeled path from poor sleep to depression is a prime example. The clinical and empirical literature strongly supports a reciprocal feedback loop where depression causes poor sleep and poor sleep exacerbates depression. The current model and its conclusions only reflect one direction of this effect. Therefore, a conclusion that the effect of poverty on depression is "fully mediated by sleep" is an oversimplification not fully supported by the data, as it does not account for this inherent bidirectionality.

Regarding question 2, while the choice of statistical methods (logistic regression, gSEM) is appropriate for the research questions, the application and reporting lack the necessary rigor.

• The presentation of the logistic regression results in Table 4 as a series of separate, unadjusted models is not ideal. A hierarchical modeling approach would more clearly demonstrate the independent contribution of various factors.

• The theoretical justification for the a priori gSEM model is underdeveloped. The specific pathways tested need a stronger grounding in existing literature or theory

My justification for the negative response to Question 3 is direct. The manuscript does not meet the journal’s data availability requirements. The authors' Data Availability Statement claims that all data are included within the article and its supplements. However, a review of these documents confirms they contain only aggregated summary statistics. The PLOS Data Policy explicitly requires that the individual, de-identified data points underlying these summary statistics (e.g., the data points behind means, medians, and variance measures) be made fully available. To comply, the authors must deposit a de-identified participant-level dataset in an appropriate public repository and include the access information in their statement.

Finally, my affirmative response to Question 4 is based on the overall quality of the writing. The manuscript is presented in an intelligible fashion and is written in standard, professional English. The narrative is logical and accessible to an expert in the field. However, to fully meet the highest standards of clarity and precision, the manuscript would benefit from a final copyedit. During the review, several important errors were noted that should be corrected:

• Inconsistencies: There are discrepancies between the results described in the text and the data presented in the tables. For example, the text reports a non-significant confidence interval for the association between being in a committed relationship and anxiety, whereas Table 4 reports a significant one. All such values must be checked meticulously.

• Table Accuracy: The frequencies in Table 1 for marital status and education doesn’t add up when manually added the frequencies for lower SES and higher SES to the total population, i.e., "Married" status do not sum correctly (3 + 51 ≠ 83). All tables should be reviewed for accuracy.

**Do you want your identity to be public for this peer review?** For information about this choice, including consent withdrawal, please see our Privacy Policy

Reviewer #1: **Yes:** Nonye Tochi Aghanya MSc, RN, FNP-C

Reviewer #2: **Yes:** Fardin Araf

---

## [Decision Letter · Decision Letter 1]

5 Jan 2026

PMEN-D-25-00305R1

Preconception mental health (Healthy Life Trajectories Initiative): Identifying factors associated with probable anxiety and depression among young women living in urban-poor South Africa

PLOS Mental Health

Dear Dr. Norris,

Thank you for submitting your manuscript to PLOS Mental Health. After careful consideration, we feel that it has merit but does not fully meet PLOS Mental Health’s publication criteria as it currently stands. Therefore, we invite you to submit a revised version of the manuscript that addresses the points raised during the review process.

We look forward to receiving your revised manuscript.

Kind regards,

Gellan Karamallah Ramadan Ahmed

Academic Editor

PLOS Mental Health

Journal Requirements:

Additional Editor Comments (if provided):

Reviewers' comments:

Reviewer's Responses to Questions

**Comments to the Author**

Reviewer #2: All comments have been addressed

Reviewer #3: All comments have been addressed

publication criteria?

Reviewer #2: No

Reviewer #3: Yes

3. Has the statistical analysis been performed appropriately and rigorously?

Reviewer #2: Yes

Reviewer #3: Yes

4. Have the authors made all data underlying the findings in their manuscript fully available (please refer to the Data Availability Statement at the start of the manuscript PDF file)?

Reviewer #2: Yes

Reviewer #3: Yes

5. Is the manuscript presented in an intelligible fashion and written in standard English?

Reviewer #2: Yes

Reviewer #3: Yes

Reviewer #2: Does this manuscript meet PLOS Mental Health’s publication criteria?

Response: Partly. Reason: The manuscript fails Criteria 3 (Technical Standard) and Criteria 4 (Conclusions supported by data) due to critical data accuracy failures. The Internal statistical/reporting inconsistencies require a full audit before the work can be considered technically sound.

There are several discrepancies and apparent errors between the results text and tables, and within tables, including:

• Results text misreports the depression OR for hazardous alcohol as 3.18 (which matches the poor sleep OR) (Manuscript PDF p.15/39 vs Table 4 Model 4, Manuscript PDF p.17/39).

• The discussion states food insecurity was “significantly associated with poorer mental health outcomes” (Manuscript PDF p.24/39), but adjusted logistic regressions show non-significant associations for food insecurity across outcomes (Table 4 Models 3–5; Table 5) (Manuscript PDF pp.17–18/39).

• Consuming a hazardous amount of alcohol... depression: OR 3.18 is mentioned in narrative. But, in Table 4, (Model 4), Alcohol OR is 1.29

• Table 4 (Anxiety Model 4) and Table 5 report identical coefficients (e.g., Sleep OR 4.03), The Given the different outcomes and sample size, this looks surprising. I would like to request the autgors to recheck this.

• Table 4 reports hazardous alcohol as highly significant for anxiety (1.09*), while the CI spans 1.00 (0.85;1.39) (Manuscript PDF p.17/39).

• Table 6 contains implausible CI values (e.g., -139 to 139; -164 to 164) for poverty→(depression/anxiety) via food insecurity (Manuscript PDF p.21/39), as well as formatting errors in at least one CI cell (Manuscript PDF p.21/39).

Why these are problems?

PLOS publication criteria require analyses to be conducted and reported to a high technical standard (Reviewer Guidelines: criterion 3; Publication Criteria: criterion 3). Inconsistencies of this type compromise interpretability and may indicate deeper reproducibility problems (e.g., incorrect model outputs, transcription errors, or mis-specified models).

Has the statistical analysis been performed appropriately and rigorously?

Response: Yes. The remaining issues are considerable. They are as follows:

1. While data missingness is acknowledged and variable n’s appear in descriptive tables (Manuscript PDF p.13/39), the manuscript does not state how missing data were handled in logistic regression and gSEM (complete-case vs imputation), nor does it provide analytic sample sizes for each model.

2. Methods specify RMSEA and BIC as fit criteria (Manuscript PDF p.13/39), but realised RMSEA/BIC are not reported. Given the centrality of gSEM in the conclusions, fit statistics should be presented.

3. Methods state sensitivity analyses across SES strata were conducted (Manuscript PDF p.13/39), but no results are presented (Not stated/unclear). Please report these.

Reviewer #3: 1. The Odds Ratio of 32.20 (95% Confidence Interval: 24.34–42.60) shows a very high link between moderate-to-severe anxiety and depression. This suggests that, in this study, anxiety and depression may be viewed as almost the same thing. The GAD-7 and PHQ-9 questionnaires share common symptoms, such as sleep problems and restlessness. The authors should clarify whether these tools are differentiating between two different conditions or just measuring general psychological distress.

2. The manuscript often uses words like "drive a syndemic," "influence," and "effects." Since the data is collected at a single point in time, we cannot determine the direction of the relationship (for example, whether Depression leads to Sleep issues or sleep issues lead to Depression). We can only make theoretical assumptions. The authors need to provide stronger statistical support for rejecting the idea that sleep and depression affect each other.

3. The study uses a household asset score. While this is a common approach, it may not effectively differentiate between the "poorest of the poor" and the "relatively stable poor" in a similar urban setting. This could limit the observed effects related to social structure.

4. In the Discussion section, likely in the Limitations part about self-reporting. It is suggested to refer from the paper: “Mental Health Monitoring and Intervention Using Unsupervised Deep Learning on EEG Data “. The manuscript notes that self-reported mental health symptoms can be influenced by social desirability bias. This reference is important because it contrasts traditional survey methods with new objective approaches. It strengthens the manuscript by highlighting that, while self-reports are useful (as seen in this study), future mental health methods may rely more on objective measures (such as EEG) and unsupervised learning to minimize misreporting and labeling issues. This places the study's methodology within the changing landscape of mental health diagnostics.

5. Check for overlap between GAD-7 and PHQ-9 scores. Perform a Confirmatory Factor Analysis (CFA) to show that these scores represent different factors in this sample. Consider whether symptom overlap (like sleep items included in both questionnaires) inflates this odds ratio. It may be useful to run the analysis without the "sleep" item in either score to see if the odds ratio changes. Revisit Model 5. Using one mental health condition as a predictor for another in logistic regression may obscure the effects of variables like socioeconomic status (SES) and adverse childhood experiences (ACEs) due to excessive modification.

6. Recognize the limits of generalized Structural Equation Modeling (gSEM) with this cross-sectional data. Replace terms like "effects" and "influences" with "associations" or "pathways" throughout the Results and Discussion sections. Clarify why the authors rejected the "Depression leads to Sleep" path in favor of "Sleep leads to Depression." Did they base this on model fit alone (using the Bayesian Information Criterion)? They need to explain their reasoning, as sleep problems are a key symptom of depression, suggesting that the causal relationship may work in both directions.

7. The following reference can improve the manuscript’s discussion on limitations, especially regarding self-reported versus objective measures, the pandemic context of the data, and future technological implications of their proposed interventions:

**Do you want your identity to be public for this peer review?** For information about this choice, including consent withdrawal, please see our Privacy Policy

Reviewer #2: **Yes:** Fardin Araf

Reviewer #3: No

---

## [Decision Letter · Decision Letter 2]

4 Mar 2026

Preconception mental health (Healthy Life Trajectories Initiative): Identifying factors associated with probable anxiety and depression among young women living in urban-poor South Africa

PMEN-D-25-00305R2

Dear Professor Norris,

We are pleased to inform you that your manuscript 'Preconception mental health (Healthy Life Trajectories Initiative): Identifying factors associated with probable anxiety and depression among young women living in urban-poor South Africa' has been provisionally accepted for publication in PLOS Mental Health.

Best regards,

Gellan Karamallah Ramadan Ahmed

Academic Editor

PLOS Mental Health

Reviewer Comments (if any, and for reference):

Reviewer's Responses to Questions

**Comments to the Author**

Reviewer #3: All comments have been addressed

Reviewer #4: All comments have been addressed

publication criteria?

Reviewer #3: Yes

Reviewer #4: Yes

3. Has the statistical analysis been performed appropriately and rigorously?

Reviewer #3: Yes

Reviewer #4: Yes

4. Have the authors made all data underlying the findings in their manuscript fully available (please refer to the Data Availability Statement at the start of the manuscript PDF file)?

Reviewer #3: Yes

Reviewer #4: Yes

5. Is the manuscript presented in an intelligible fashion and written in standard English?

Reviewer #3: (No Response)

Reviewer #4: Yes

Reviewer #3: I recommend accepting this revised manuscript. The authors have carefully addressed all the points raised in the previous review with great attention to detail. They have resolved every concern regarding statistical interpretation, model specification, and clarity of writing through effective revisions. The current version of the paper is strong, organized, and ready for publication.

1. The authors provide a clear response resolving previous concerns, including correcting errors and aligning findings with data.

2. The manuscript shows improved clarity and organization, especially in the methodology with sample sizes for each model.

3. The technical quality is enhanced by adding a confirmatory factor analysis confirming distinct constructs.

4. The paper timely contributes to global mental health by exploring socioeconomic stress and early-life trauma among young women in under-resourced South African urban areas.

5. Analysis is strengthened by re-evaluating the gSEM with a multinomial food insecurity variable, improving fit and confidence intervals.

6. Overall presentation improved by moderating causal language and discussing limitations, like asset scores and the cross-sectional design.

Reviewer #4: I have read the paper thoroughly and the reviewer's suggestions were addressed properly as per my perception.

Good Luck

**Do you want your identity to be public for this peer review?** For information about this choice, including consent withdrawal, please see our Privacy Policy

Reviewer #3: No

Reviewer #4: No
